# Change in Pediatric Health Care Spending and Drug Utilization during the COVID-19 Pandemic

**DOI:** 10.3390/children8121183

**Published:** 2021-12-15

**Authors:** Riccardo Lubrano, Emanuela Del Giudice, Alessia Marcellino, Flavia Ventriglia, Anna Dilillo, Enrica De Luca, Saverio Mallardo, Sara Isoldi, Vanessa Martucci, Mariateresa Sanseviero, Donatella Iorfida, Concetta Malvaso, Giovanni Cerimoniale, Giuseppina Ragni, Anna Lisa Grandinetti, Loredana Arenare, Silvia Bloise

**Affiliations:** 1Dipartimento Materno Infantile e di Scienze Urologiche, Sapienza Università di Roma, UOC di Pediatria e Neonatologia–Polo Pontino, 04100 Latina, Italy; Riccardo.lubrano@uniroma1.it (R.L.); emanuela.delgiudice@gmail.com (E.D.G.); marcellino.alessia@gmail.com (A.M.); flavia.ventriglia@uniroma1.it (F.V.); Annadilillo83@gmail.com (A.D.); enrideluca@live.it (E.D.L.); saverio.mallardo@gmail.com (S.M.); isoldi.sara@gmail.com (S.I.); vany.mart@gmail.com (V.M.); mariateresa.sanseviero@yahoo.it (M.S.); donatella.iorfida@gmail.com (D.I.); 2Pediatri di Famiglia Azienda Sanitaria Locale di Latina, 04100 Latina, Italy; cettimalvaso@libero.it (C.M.); giovanni.cerimoniale@alice.it (G.C.); giuseppinaragni@gmail.com (G.R.); 3Direzione Sanitaria Azienda Sanitaria Locale di Latina, 04100 Latina, Italy; a.grandinetti@ausl.latina.it; 4UOC Farmaceutica Territoriale e Integrativa Azienda Sanitaria Locale di Latina, 04100 Latina, Italy; l.arenare@ausl.latina.it

**Keywords:** preventive measures, children, infectious diseases, drugs, pediatric and healthcare, expenditure

## Abstract

Objective: To evaluate how the restrictive measures implemented during the SARS-CoV-2 pandemic have influenced the incidence of the most common children’s diseases and the consumption of medications in 2020 compared to 2019. Methods: We involved all family pediatricians of the local health authority of Latina, from which we requested data of monthly visits in 2019 and 2020 for six common diseases disseminated through droplets and contact, and the territorial and integrative pharmaceutical unit of the area, from which we requested data of the net expenditure regarding the most commonly used drugs at pediatric age. Results: There was significant reduction in the incidence of the evaluated diseases and in the consumption of investigated drugs between 2019 and 2020 in the months when the restrictive measures were in place, with an attenuation of this effect during the months of the gradual loosening of those measures. Conclusion: Nonpharmaceutical intervention measures have caused changes in the diffusion of common pediatric diseases. We believe that the implementation of a reasonable containment strategy, even outside of the pandemic, could positively influence the epidemiology of infectious and allergic diseases in children, and healthcare system spending.

## 1. Introduction

Since the end of 2019, severe acute respiratory syndrome coronavirus-2 (SARS-CoV-2), first identified in Wuhan, China, and responsible for the outbreak of severe respiratory disease, novel coronavirus disease 2019 (COVID-19), has spread rapidly around the world. On 11 March 2020, the World Health Organization (WHO) declared COVID-19 a pandemic [1].

Italy was one of the most affected countries in the world by SARS-CoV-2, with most of the cases registered in the north. In order to face the pandemic, the Italian government and public health agencies have developed and implemented several recommendations to contain the spread of infection.

Learning from the past, the main proposed preventive measures were: stay-at-home orders, social distancing, handwashing and mask wearing, not only in hospital settings, but also for the general population [2,3].

In particular, in Italy, from 9 March to 3 May 2020, the Italian prime minister declared a lockdown with the suspension of common commercial activities and catering services, the closure of schools, the prohibition of people gathering in public places, the banning of travel to municipalities other than those to which people belonged, and the introduction of the obligation to wear a facial mask, both indoors and outdoors [4,5].

Subsequently, since the epidemic curve was in a downward phase, there was a gradual loosening of previous containment measures with the reopening of most activities and the permission to travel until October 2020, when the second wave of infection [6] led to new restrictive measures being implemented from 24 October to 26 April 2021 [7].

In this context, many reports showed that the COVID-19 pandemic and the implementation of mitigation measures caused a new and unexpected scenario, both in adult [8] and pediatric care [9]. In pediatric care, there was a drastic decrease in admissions, outpatient visits, and emergency room access during the lockdown [10,11], characterized by a significant reduction in pediatric infectious diseases disseminated through droplets and contact [12,13,14].

Therefore, in this study, we draw a picture of what happened in the pediatric setting in the Latina Local Health Authority (ASL) during the 2020 infectious season, comparing it with the pre-COVID-19 era.

Our primary aim was to evaluate how restrictive measures implemented because of the SARS-CoV-2 pandemic have influenced the spread and incidence of the most common allergic and infectious diseases in 0–14-year-old children.

Our secondary aim was to assess whether this resulted in a difference in pharmaceutical expenditure of commonly used medications in the pediatric field.

## 2. Methods

This is an observational retrospective study conducted by the Pediatric Unit of Santa Maria Goretti Hospital, Latina, Sapienza University of Rome (Polo Pontino). Data collection was performed between January 2019 and December 2020; data processing was performed between January 2021 and February 2021.

The institutional review board of the Maternal and Child Health Department of the local health authority of Latina approved the study protocol.

This report follows the STROBE reporting guideline for observational studies [15].

Informed consent was obtained from all study participants.

To conduct the study, we referred to the local health authority of Latina (ASL of Latina).

In Italy, healthcare is provided free of charge by the national health service through health companies that provide care through a hospital and territorial structure. The territorial structure is formed by family doctors. Every child from 0 to 14 years of age is guaranteed assistance with a pediatrician who is freely chosen by the family and can assist a maximum of 800–1000 children.

In addition, each healthcare company is equipped with its own administrative systems of control, such as the evaluation of pharmaceutical expenditure.

The ASL of Latina represents an area corresponding to about 1/3 of the Lazio region with a total number of 75,360 pediatric patients.

We involved all family pediatricians of the area, engaged via email invitations to participate in the study, and the Territorial and Integrative Pharmaceutical Unit of the ASL of Latina to obtain the incidence of six major pediatric allergic and infectious diseases, and the extent of pharmaceutical expenditure of the most common drugs used in the pediatric age group (0–14 years); our search was related to the months of 2019 preceding the outbreak of the pandemic, and to the months of 2020 during which, following the spread of the coronavirus pandemic, two lockdown periods were implemented by the Italian government (9 March–3 May 2020; 24 October 2020–26 April 2021), characterized by nonpharmaceutical intervention measures (stay-at-home orders, social distancing, travel restrictions, handwashing, and mask wearing) to control the spread of the pandemic.

We evaluated six major pediatric allergic and infectious diseases disseminated through droplet and contact: bronchiolitis, asthma, pneumonia, laryngitis, otitis and pharyngitis.

Bronchiolitis was defined as the first lower respiratory tract infection in children less than 1 year old exposed to people presenting with upper respiratory tract viral infections or during epidemic season [16]. Asthma was defined by a history of typical clinical symptoms and signs such as wheezing, dyspnea, chest tightness and/or cough, modifiable in time and intensity, associated with a reversible airflow obstruction [17];

Pneumonia was defined as the presence of fever (>37.5) and acute respiratory symptoms (cough, tachypnea, retractions, chest pain, rales) associated with a new pulmonary infiltrate documented on chest radiography or clinical examination [18,19]. Laryngitis was defined as upper airway inflammation characterized by hoarseness, barking cough and inspiratory stridor [20]. Otitis media was defined as the presence of moderate–severe bulging of the tympanic membrane or new onset otorrhea, or by the presence of mild bulging of the tympanic membrane and <48 h onset of ear pain or intense erythema of tympanic membrane [21]. Pharingytis was defined as inflammation of the tonsils and pharynx characterized by symptoms such sore throat with or without fever with pharyngeal erythema and/or exudates on physical examination [22].

The five commonly used pediatric medications evaluated were: salbutamol (also known as albuterol), cephalosporins (cefixime, ceftibutene, cefpodoxime, ceftriaxone), amoxicillin–clavulanic acid, macrolides (clarithromycin and azithromycin) and inhaled corticosteroids (beclomethasone, fluticasone, budesonide).

Exclusion criteria were well-child visits or visits for other conditions not subject to examination (urological, traumatic, neurological, gastrointestinal).

For the primary aim of our study, we requested family pediatricians to send to us related data, coded through a software (Junior Bit-So.Se.Pe, Padova, Italy) with which all family pediatricians’ offices are currently equipped, of the visits made monthly in the years 2019 and 2020 for the diseases under investigation.

For the secondary aim related to the consumption of drugs, we requested the Territorial and Integrative Local Pharmaceutical Unit to send us data related to the net expenditure, expressed in euro (EUR), that is, the price of the drug net of pharmacy fees and patient co-payment, of the most common drugs used at pediatric age. These data were extracted from the regional Pentaho Data Warehouse and refer to all prescriptions filled by family pediatricians in the months of the years 2019 and 2020.

### Statistical Analysis

For statistical analysis, we relied on the JMP 15.2.1 program for Mac by SAS Institute Inc. For all the parameters considered in the study, the approximation to normal of the distribution of the population was tested with the Anderson–Darling test. As results were asymmetrically distributed (bronchiolitis (*p* < 0.0001), asthma (*p* < 0.0001), laryngitis (*p* < 0.0001), pneumoniae (*p* < 0.0001), otitis (*p* < 0.0001), pharyngitis (*p* < 0.0001), salbutamol (*p* < 0.05) cephalosporins (*p* < 0.002), amoxicillin–clavulanic acid (*p* < 0.008), macrolides (*p* < 0.05) and inhaled corticosteroids (*p* < 0.05)), data are expressed as median and interquartile range (IQR), 25th and 75th quartile, and non-parametric tests were used for statistical analysis. We employed the Wilcoxon test to compare the differences between the studied groups. A *p* < 0.05 was considered significant.

## 3. Results

Of 70 family pediatricians of the local health authority of Latina, 33 accepted our invitation to participate in the investigation for a total of 35,787 children involved.

### 3.1. Differences in the Incidence of Communicable Respiratory Diseases between 2019 and 2020

Table 1, Table 2, Table 3, Table 4, Table 5 and Table 6 summarize the differences in the incidence of communicable respiratory diseases recorded monthly between 2019 and 2020, and in particular bronchiolitis, asthma, laryngitis, pneumoniae, otitis and pharyngitis.

In particular, we found a significative reduction in the incidence of all the assessed diseases during the lockdown period. The reduction was observed also in the months immediately following the lockdown, which were characterized by a gradual loosening of previous containment measures, with slight attenuation in August, when restrictive measures were no longer in force and social relations increased during summer vacations.

### 3.2. Differences in Net Expenditure of Common Pediatric Drugs between 2019 and 2020

We also found a significative difference in net expenditure between 2019 and 2020 for all drugs considered.

There was a significative reduction in the annual median of net expenditure for salbutamol (493 (303–1421) vs. 1959 (1227–2723), *p* < 0.0226), cephalosporin (4006 (2381–6786) vs. 13,073 (10,096–19,130), *p* < 0.0086), amoxicillin–clavulanic acid (6035 (3795–11,897) vs. 20,280 (14,347–31,278), *p* < 0.0061), macrolides (2652 (1508–6811) vs. 9857 (7437–14,726), *p* < 0.0120) and inhaled corticosteroids (7187 (3826–19,126) vs. 25,549 (12,202–30,312), *p* < 0.0464).

In addition, Figure 1, Figure 2, Figure 3, Figure 4 and Figure 5 show the trend in net expenditure and delta 2020/2019 of net monthly expenditure for each of the drugs considered, highlighting the drastic reduction in the consumption of these medications in 2020.

It is interesting to note how the trend in pharmaceutical expenditure varied in relation to the implementation of containment measures; in particular, it was higher in the first two months of 2020 compared to 2019, when the containment measures were not in place, and fell dramatically during the lockdown periods, with a slight increase in August, when restrictive measures were suspended, until a new reduction during the second lockdown.

## 4. Discussion

Our results show that the year 2020, dominated by COVID-19, was a year unlike any other, with significant changes in children’s pediatric care.

Primarily, along with a non-severe spread of SARS-CoV-2 infection in children [23,24], we observed a drastic reduction in the incidence of the main pediatric diseases caused by atopy and common seasonal bacteria and viruses. Not surprisingly, in 2020 the typical increase in consultations and hospital admissions for respiratory syncytial virus (RSV) and influenza viruses did not occur during the winter season [13,25,26]. Furthermore, emergency consultations for air communicable diseases were less than that for other conditions, such as urinary infections, neurological and surgical problems or accidents [12,27,28,29].

Secondarily, in line with the reduction in pediatric infections and allergies during 2020, a significant reduction in the consumption of commonly prescribed pediatric medication was observed. In particular, we found a significant reduction in the prescription of all drug classes considered, such as bronchodilators and inhaled corticosteroids and antibiotics with different spectra of action. The decrease in allergic diseases and asthma was previously suggested by other studies [30,31,32,33].

The main reasons for this new scenario seem to be related to the implementation or the mitigation of the measures taken to reduce the spread of SARS-CoV-2, which indirectly reduced the circulation and transmission of other bacteria and respiratory viruses. As proof of this, previous studies showed the effectiveness of these strategies in the control of epidemics [34,35,36,37,38,39,40].

Our study supports this hypothesis; in fact, the differences in the incidence of pediatric infections and allergies are particularly evident during the lockdown period.

In particular, our data reflect the epidemiologic change in common pediatric diseases that occurred during the study period: the peak of infants with bronchiolitis that is generally recorded 2–3 months after the beginning of the cold season was here missed, and the high incidence of laryngitis cases in the fall period did not occur; furthermore, we did not witness the asthma exacerbations triggered by respiratory viral infections and outdoors allergens typical of the winter and spring periods.

A strength of our study was that we collected data from two independent organizations of the ASL of Latina. We analyzed the data of the family pediatricians for the epidemiological analysis of six common childhood diseases, and at the same time, we evaluated the corresponding pharmaceutical expenditure from questioning the territorial and integrative pharmaceutical unit, drawing the same conclusions. Our results are the expression of the same phenomenon and allow us to depict some changes that occurred in the pediatric field in the COVID-19 era.

Therefore, while the COVID-19 pandemic has strained our health care system, it has indirectly opened up new perspectives on the importance of infection prevention through control measures that could impact the dynamics of various allergic and infectious diseases and that could positively influence the burden and spending of our health care system during epidemic seasons [41,42,43,44,45].

Each of the preventive measures adopted produced a positive effect: social distance, orders to stay home and travel restrictions prevented direct person-to-person contact; facial masking, creating a physical barrier in front of the nose and mouth, and continuous sanitization of contact surfaces contributed to the reduction in the spread of respiratory viruses and of the penetration of allergens and environmental pollutants that are important bronchial irritants [46,47,48]; and hand washing and the use of disinfectants acted on enveloped RNA viruses particularly sensitive to soaps and detergents [49], causing the reduction in the transmission of infective agents.

However, such rigid containment measures also led to negative psychological effects, especially in children; in fact, the closure of schools, parks and sporting activities and the orders to stay at home have disrupted children’s usual lifestyle, promoting distress, confusion, anxiety and hostility, with possible relevant effects on their psychosocial well-being and cultural education in the long term [50,51,52]. In particular, many parents have questioned the use of face masks by their children, raising concerns about possible negative effects on respiratory function that are not supported by proven scientific evidence [45,53,54].

Therefore, we believe that it is necessary to achieve a balance in the adoption of non-pharmaceutical intervention measures during pandemics, not secondary to SARS-CoV-2 infection; in fact, the rational use of mitigation measures, such as an appropriate use of masks during periods of seasonal infectivity controlling the variolation phenomenon [55] or during the pollination period, could lead to a drastic reduction in the incidence of infectious and allergic diseases in children.

As demonstrated by the recent outbreak, these measures could therefore contribute to a reduction in pharmaceutical and hospital expenditure, with an indirect reduction in social expenditure related to the absenteeism from work of the parents of affected children.

## Figures and Tables

**Figure 1 children-08-01183-f001:**
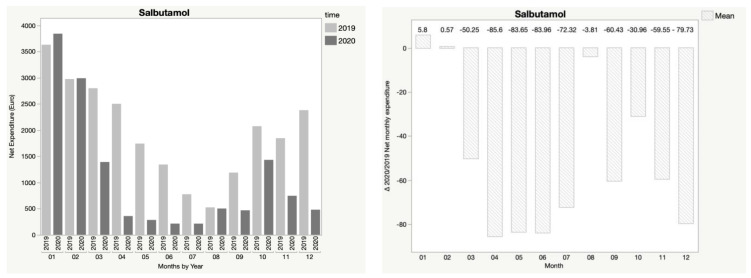
Trend in net expenditure (EUR) and delta 2020/2019 of net monthly expenditure of salbutamol, from January through December of the years 2019 and 2020, in the study population of 35,787 children aged 0–14 years.

**Figure 2 children-08-01183-f002:**
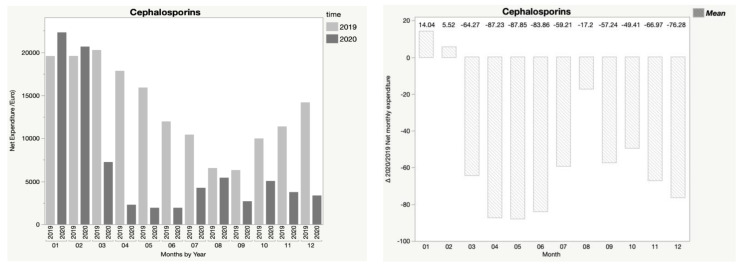
Trend in net expenditure (EUR) and delta 2020/2019 of net monthly expenditure of cephalosporin from January through December of the years 2019 and 2020, in the study population of 35,787 children aged 0–14 years.

**Figure 3 children-08-01183-f003:**
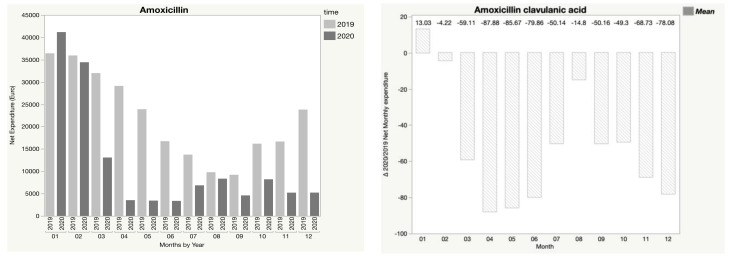
Trend in net expenditure (EUR) and delta 2020/2019 of net monthly expenditure of amoxicillin–clavulanic acid, from January through December of the years 2019 and 2020, in the study population of 35,787 children aged 0–14 years.

**Figure 4 children-08-01183-f004:**
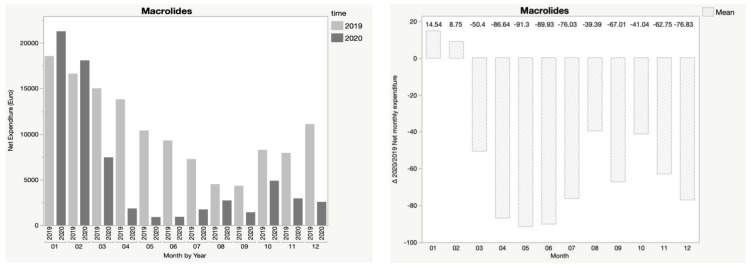
Trend in net expenditure (EUR) and delta 2020/2019 of net monthly expenditure of macrolides, from January through December of the years 2019 and 2020, in the study population of 35,787 children aged 0–14 years.

**Figure 5 children-08-01183-f005:**
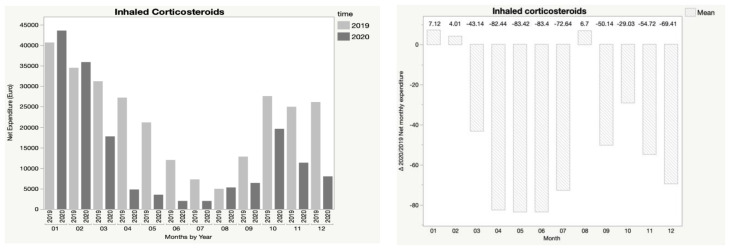
Trend in net expenditure (EUR) and delta 2020/2019 of net monthly expenditure of inhaled corticosteroids, from January through December of the years 2019 and 2020, in the study population of 35,787 children aged 0–14 years.

**Table 1 children-08-01183-t001:** Number of patients with bronchiolitis recorded monthly in 2019 compared to 2020, in the study population of 35,787 children aged 0–14 years.

Months	2019 Median (IQR)	2020 Median (IQR)	*p*
January	2.5 (0–6.25)	2 (0–7)	0.80
February	3 (1–8)	2 (0–6)	0.34
Start of first lockdown			
March	1 (0–2)	0 (0–1)	0.0136
April	0.5 (0–3.25)	0 (0–0)	0.0010
May	0 (0–1)	0 (0–0)	0.0065
End of first lockdown			
June	0 (0–0)	0 (0–0)	0.418
July	0 (0–0)	0 (0–0)	0.557
August	0 (0–0)	0 (0–0)	1
September	0 (0–0)	0 (0–0)	0.483
October	0 (0–1)	0 (0–0)	0.192
Start of second lockdown			
November	0 (0–3.25)	0 (0–0)	0.0079
December	1 (0–2.5)	0 (0–0)	0.0019

**Table 2 children-08-01183-t002:** Number of patients with asthma recorded monthly in 2019 compared to 2020, in the study population of 35,787 children aged 0–14 years.

Months	2019 Median (IQR)	2020 Median (IQR)	*p*
January	8.5 (4–15.5)	6 (1.75–18.25)	0.45
February	5.5 (2–16.5)	5 (1.75–17.25)	0.70
Start of first lockdown			
March	7 (2.75–13.25)	0 (0–4.25)	0.0006
April	6 (3.75–11.25)	0.5 (0–2)	<0.0001
May	5 (2–9.25)	1 (0–2.25)	<0.0001
End of first lockdown			
June	2.5 (1–6.5)	0 (0–2)	0.0002
July	1 (0–4)	0 (0–2)	0.1253
August	0 (0–1)	0 (0–0)	0.1652
September	2 (0–4.25)	0 (0–2)	0.0235
October	4 (0–6)	1 (0–4)	0.060
Start of second lockdown			
November	2 (1–8.5)	0.5 (0–3.75)	0.0295
December	4 (1–9.25)	0 (0–2.25)	0.0005

**Table 3 children-08-01183-t003:** Number of patients with pneumonia recorded monthly in 2019 compared to 2020, in the study population of 35,787 children aged 0–14 years.

Months	2019 Median (IQR)	2020 Median (IQR)	*p*
January	14 (3–23.5)	16.5 (8–28)	0.39
February	14.5 (5.75–23)	12 (6–25.25)	0.86
Start of first lockdown			
March	8 (5–20.75)	2 (0.75–4.25)	<0.0001
April	8.5 (5.75–15.25)	0 (0–0)	<0.0001
May	5 (2–13)	0 (0–1)	<0.0001
End of first lockdown			
June	3.5 (1–7.25)	0 (0–0)	<0.0001
July	2 (0–4)	0 (0–0)	0.0002
August	0 (0–1.25)	0 (0–0)	0.049
September	2 (0–3.25)	0 (0–2)	0.055
October	4.5 (1–10)	2 (0–3)	0.052
Start of second lockdown			
November	5 (2–15)	1 (0–2)	0.0002
December	6 (4–16.5)	0.74 (0–4.25)	<0.0001

**Table 4 children-08-01183-t004:** Number of patients with laryngitis recorded monthly in 2019 compared to 2020, in the study population of 35,787 children aged 0–14 years.

Months	2019 Median (IQR)	2020 Median (IQR)	*p*
January	2.5 (1–14.5)	4 (1.75–12)	0.70
February	4 (2–14.25)	2.5 (0–10.25)	0.29
Start of first lockdown			
March	4 (1–11)	0 (0–3)	0.0005
April	2.5 (0.75–5.25)	0 (0–0)	<0.0001
May	1.5 (0.75–4)	0 (0–0)	<0.0001
End of first lockdown			
June	1 (0–2)	0 (0–0)	<0.0021
July	0 (0–1)	0 (0–0)	0.123
August	0 (0–1)	0 (0–0.25)	0.370
September	1 (0–4)	0 (0–1)	0.0058
October	3 (0–8.5)	1 (0–3)	0.0185
Start of second lockdown			
November	2 (0–10)	0 (0–2)	0.0053
December	2.5 (1–7)	0.5 (0–3)	0.0118

**Table 5 children-08-01183-t005:** Number of patients with otitis recorded monthly in 2019 compared to 2020, in the study population of 35,787 children aged 0–14 years.

Months	2019 Median (IQR)	2020 Median (IQR)	*p*
January	10 (6–20.5)	10.5 (5–21)	0.98
February	15.5 (7.75–24.75)	10 (5.5–20.75)	0.24
Start of first lockdown			
March	12.5 (6.75–25.75)	3 (0–5)	<0.0001
April	8 (4–15.25)	0 (0–1.25)	<0.0001
May	8.5 (4–12.75)	0 (0–1)	<0.0001
End of first lockdown			
June	5 (2.75–10.25)	0.5 (0–2)	<0.0001
July	7.5 (3–16.25)	3.5 (1–8)	0.0274
August	4 (2–7.25)	4 (0–8.25)	0.49
September	3.5 (1.75–8.25)	1 (0–3.25)	0.0204
October	6.5 (2–11.25)	3 (0–5.25)	0.044
Start of second lockdown			
November	7 (2.75–14.5)	2 (0–4.25)	0.0063
December	9 (4.75–15)	2 (0–5.5)	<0.0001

**Table 6 children-08-01183-t006:** Number of patients with pharyngitis recorded monthly in 2019 compared to 2020, in the study population of 35,787 children aged 0–14 years.

Months	2019 Median (IQR)	2020 Median (IQR)	*p*
January	25.5 (18–47)	26 (14.25–60.75)	0.97
February	29 (17.75–36.5)	19 (7.5–44)	0.29
Start of first lockdown			
March	29 (14–41.25)	3.5 (0–6.25)	<0.0001
April	28 (10.75–43.25)	0 (0–3.25)	<0.0001
May	18.5 (7–32)	1 (0–4.25)	<0.0001
End of first lockdown			
June	14.5 (4.75–27.25)	1 (0–2.25)	<0.0001
July	9 (2.75–15.75)	1 (0–2.25)	<0.0001
August	4 (1.5–12)	2.5 (0–6.25)	0.14
September	10.5 (3–16.5)	2 (0–5.25)	0.0010
October	16 (5.75–30.5)	4 (1.75–12)	0.0014
Start of second lockdown			
November	17.5 (5.75–31.5)	3 (1.75–12.25)	0.0003
December	20 (9–36.75)	2 (0.75–12)	<0.0001

## Data Availability

All data and materials support published claims and comply with field standards.

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
