# Peer review of "Change in Pediatric Health Care Spending and Drug Utilization during the COVID-19 Pandemic"

_children, 2021, doi:10.3390/children8121183_

Round 1
Reviewer 1 Report
The present paper provides scientific grounds for a phenomenon that was widely observed by family doctors and pulmonologists across the world during the COVID-19 pandemic and is still ongoing. That the respiratory infections and allergic conditions significantly dropped since the compulsory measures were put in place, namely disinfection and social distancing. The article has excellent introduction describing in clear and yet very concise manner the background, the use of two databases serves as control method to prove the validity of the results. Results are very well presented and conclusions drawn in the discussions section follow them. English language is of high level, only additional spellcheck would be necessary. There are several extra spaces in the text and on line "112" throat is misspelled as "troth". Thank you!Author Response
Please see the attachment

Reviewer 2 Report
Major comments:
- Needs English language edits
- Did the study population include children who suffered from COVID-19? Can the authors please clarify?
- Any exclusion criteria applied on the study population?
- In the methods section, study period described as January 2021 to February 2021 however data collection dates to a future date i.e. 26 April 2021. Could the authors kindly clarify this?
- In the methods section, consider adding information on how the data gathered from all participating physicians was centrally curated. Name the software/application used.
- Differences of net expenditure (line 167): amoxicillin clavulanic acid [6035(303 – 1421) vs 1959 (1227 – 2723) p < 0.0226.
Please check for typographical errors. Is it 1959 vs. 6035?
- Salbutamol; consider adding that this drug is also known as Albuterol.
- Include the names of cephalosporins, macrolides and inhaled corticosteroids included in the study.
Specific comments:
Title: Consider rephrasing as ‘ Change in pediatric health care spending and drug utilization during the COVID-19 pandemic’ OR ‘Change in pediatric health care spending and pharmaceutical utilization during the COVID-19 pandemic.’
Keywords: Consider adding the words: pediatric and healthcare expenditure.
Introduction:
Line 33: Define SARS-Cov-2
Line 33: Wuhan city, à Wuhan, China
Line 36: Define WHO
Methods:
Line 90: pediatric age à pediatric age group (0-14 years)
Lines 93/94: (9 March - 3 May; 24 October - 26 April)
Please add the year(s) 2020, 2021 where applicable
Line 117: coded through a software
Consider adding software name
Results:
Lines 140, 141: “air communicable diseases”
Consider rephrasing as ‘communicable respiratory diseases’
Line 165: significative à significant
Discussion:
Line 201: in this year à Could the authors kindly clarify which year 2020 or 2021?
Line 201: caused by allergies à caused by atopy
Line 207: Consider adding a reference
Line 222: “pediatric infections e allergies”
Please correct the typographical error.
Lines 223-227: Could the authors add graphical representation of the presented information?
Line 234: the use of 5 drugs used à the categories of drugs used/classes of drugs used
Line 235: spy of a common behaviour: Unclear what this means; consider editing.
Line 251: rigid containment measures have also caused à have led to
Line 253: Please spell check the word disrupted
